# Van der Waals lattice-induced colossal magnetoresistance in $Cr_2Ge_2Te_6$ thin flakes

Wenxuan Zhu[1], Cheng Song [1]✉, Lei Han[1], Tingwen Guo[1], Hua Bai[1] & Feng Pan[1]✉

Recent discovery of two-dimensional (2D) magnets with van der Waals (vdW) gapped layered structure prospers the fundamental research of magnetism and advances the miniaturization of spintronics. Due to their unique lattice anisotropy, their band structure has the potential to be dramatically modulated by the spin configuration even in thin flakes, which is still unexplored. Here, we demonstrate the vdW lattice-induced spin modulation of band structure in thin flakes of vdW semiconductor $Cr_2Ge_2Te_6$ (CGT) through the measurement of magnetoresistance (MR). The significant anisotropic lattice constructed by the interlayer vdW force and intralayer covalent bond induces anisotropic spin-orbit field, resulting in the spin orientation-dependent band splitting. Consequently, giant variation of resistance is induced between the magnetization aligned along in-plane and out-of-plane directions. Based on this, a colossal MR beyond 1000% was realized in lateral nonlocal devices with CGT acting as a magneto switch. Our finding provides a unique feature for the vdW magnets and would advance its applications in spintronics.

Discovery of intrinsic 2D vdW magnets[1-4] demonstrates the existence of magnetism in the 2D limit and supplements a new functional dimension in vdW materials which shows potential applications in magnetic memories and magnetoresistance sensors[5-7]. Besides the reproduction of abundant spintronic phenomena in vdW magnets, such as spin-orbit torque[8,9] and skyrmions[10-13], the unique crystal structure with interlayer vdW gaps makes 2D vdW magnetic materials attractive in both applicative spintronics and fundamental researches. For spintronic devices, the easy exfoliation and assembly features of 2D vdW magnets provide atomically sharp interface, giving birth to a novel category of the spinterface[14,15], the significant electrical control of magnetism[16-18], and the combination of multiple physical properties[19]. Additionally, the vdW magnets are effective spin filter and spin carrier which produces extremely large tunneling magnetoresistance (TMR) effects in vdW tunnel junctions[20-22]. For fundamental researches, the vdW gapped structure evokes a new dimension of interlayer stacking order for the manipulation of magnetism[23-25]. The weak interlayer exchange coupling also induces abnormal behavior in interfacial effects, including the exchange bias[26] and interfacial enhancement of magnetism[27]. These inspiring results benefit intrinsically from the interlayer vdW force induced gapped structure.

Actually, when further considering the intralayer covalent bond, the resultant significant anisotropy between the interlayer and intralayer lattice brings about the potential for exotic magnetic phenomena, which remains unexplored.

The different interlayer and intralayer lattice results in much weaker interlayer exchange coupling compared to the intralayer in 2D vdW magnets as shown in Fig. 1a, leading to the extension of orbital along the out-of-plane direction. Consequently, the anisotropic effective spin-orbit field $H_{SOC}$ proportional to the orbital momentum $L$ is induced. With the magnetization along the out-of-plane direction, the bands of opposite spin components significantly split with $\Delta E \sim H_{SOC} \cdot M$. In contrast, the band splitting is much weaker with in-plane magnetization. Therefore, with the unique anisotropic lattice, the band structure of 2D vdW magnets can be manipulated effectively through the spin orientation, which is named as the magneto band-structure (MB) effect[28,29]. Recently, the MB effect was observed in bulk crystal materials due to topological nodal-line bands[30,31] and symmetry-broken correlated band reconstruction[32], resulting in colossal MR effect with different spin orientations. Compared to those bulk crystal materials, 2D vdW magnets possess the advantage of being easily exfoliated from crystals and transferred with the preservation of fine

[1]Key Laboratory of Advanced Materials, School of Materials Science and Engineering, Beijing Innovation Center for Future Chips, Tsinghua University, Beijing 100084, China. ✉e-mail: songcheng@mail.tsinghua.edu.cn; panf@mail.tsinghua.edu.cn

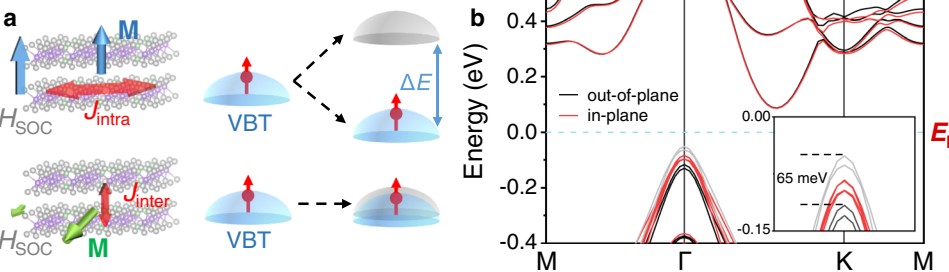

**Fig. 1 | Van der Waals lattice-induced magneto band-structure effect in CGT.** **a** Schematic of the band splitting induced by the interaction between the anisotropic crystalline filed and the magnetization. The top of valance bands (VBT) occupied by spin majority and minority are colored in blue and grey, respectively. **b** Calculated band structures of CGT bilayer with the in-plane and out-of-plane magnetizations. The fermi level is set to zero. The bands of spin majority and minority with out-of-plane magnetization at the top of the valance bands are colored in black and grey, respectively. The magnified bands below fermi level are shown in the inset.

crystal quality. This enables the efficient modulation of band structures by the magnetization even in thin flakes and heterostructures, which is demanded in highly integrated electronic devices. Furthermore, the origin of anisotropic atomic alignment inherent to vdW layered structure makes the MB effect promising to be widely observed in 2D vdW magnets.

In this work, through the measurements of MR, we experimentally demonstrated the vdW lattice-induced MB effect in thin flakes of vdW ferromagnetic semiconductor $Cr_2Ge_2Te_6$ (CGT). A significant resistance change was generated between the magnetization along in-plane and out-of-plane directions, which is much larger than the anisotropic magnetoresistance (AMR) in conventional magnetic thin films. Based on the giant resistance change, the colossal MR beyond 1000% was realized in the lateral nonlocal device with the channel of CGT which is applicable to the integration with various dimensions. The illumination of our work not only reveals the fundamental properties of vdW magnets but also advances their potential applications in the electronic devices.

## VdW lattice induce band modulation in $Cr_2Ge_2Te_6$

CGT is a typical 2D magnetic semiconductor, which possesses both interlayer and intralayer ferromagnetic coupling with Curie temperature ($T_C$) around 70 K[1]. The choice of CGT is on account of the semiconductor band structure, which will markedly influence the electrical properties after its modulation. With significant lattice anisotropy between interlayer and intralayer, the band structure of CGT can be modulated by the spin orientation as schematically illustrated in Fig. 1a. The Te-$p$ orbitals extend along the out-of-plane and the top of the valance bands occupied by Te-$p$ orbital splits due to $H_{SOC}$ which is parallel to the magnetization **M**. And the spin majority and minority occupy the bands with lower and higher energy, respectively. Due to the anisotropic $H_{SOC}$, the band splitting at the top of the valance bands can be modulated by the spin orientation. Consequently, with the hole-carrier in CGT (Supplementary Note 1), the modulation at the top of the valance bands will largely cause the change of resistance, leading to MR effect. To investigate the effect of band splitting on the electrical transport, band structures of CGT bilayer with in-plane and out-of-plane **M** are calculated. As exhibited in Fig. 1b, the splitting at top of the valance bands is quite significant with out-of-plane **M** and $\Delta E = 65$ meV compared to the in-plane **M** with $\Delta E = 0.7$ meV. The indirect band gap remains in both situations. With the large band splitting of out-of-plane **M**, the energy of band occupied by spin majority decreases which widens its band gap. In ferromagnetic CGT, the carrier is highly spin-polarized. The spin minority will be scattered rapidly and contribute little to the electric conduction. Consequently, compared to the in-plane **M**, the conductivity of semiconductor CGT, which is dominated by the spin majority, is reduced with larger resistance. Therefore, the

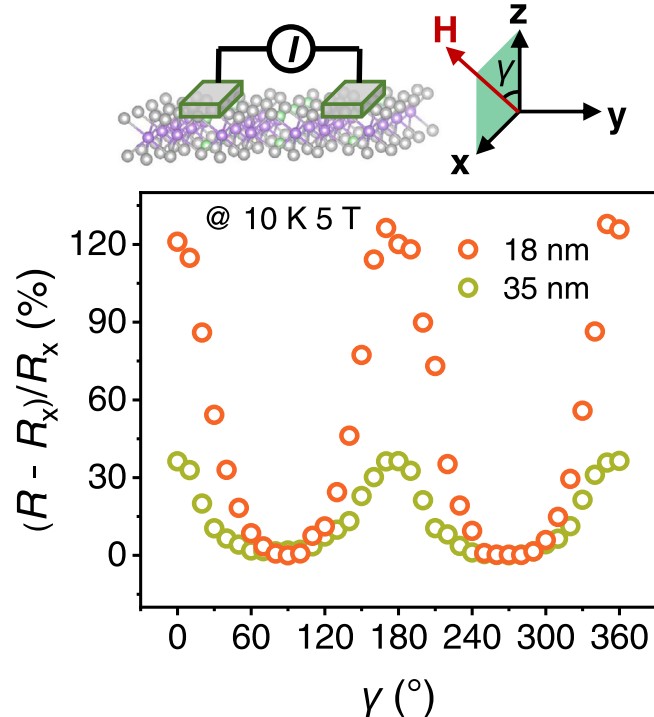

**Fig. 2 | Local MR effect induced by giant MB effect.** The insets show the schematic and the definition of azimuthal angles and directions of the measurement.

anisotropic lattice can induce large MR effect with different spin orientations.

Lattice-induced MR is then verified by measurements of local angular MR with the current directly applied in the channel of CGT between two electrons along $y$ direction as schematically shown in the inset of Fig. 2. To exclude the contribution of the conventional AMR effect, the magnetization was rotated in the plane perpendicular to the current ($\gamma$) with the applied magnetic field of 5 T at 10 K. In Fig. 2, the angle-dependent MR (defined as $(R - R_x (\gamma = 90°)) / R_x$) caused by vdW lattice is quite significant compared to conventional AMR, which reaches beyond 100% in the thin flake of 18 nm. The change of resistance, which results from the vdW lattice-induced modulation of band structure in CGT, is consequently consistent with the theory proposed above. Compared to the sample of 35 nm, the further enhancement in the thinner sample shows the characteristic of 2D materials and reflects its origin of the vdW layered structure. This is supported by the calculations of thickness-dependent band structure which shows the

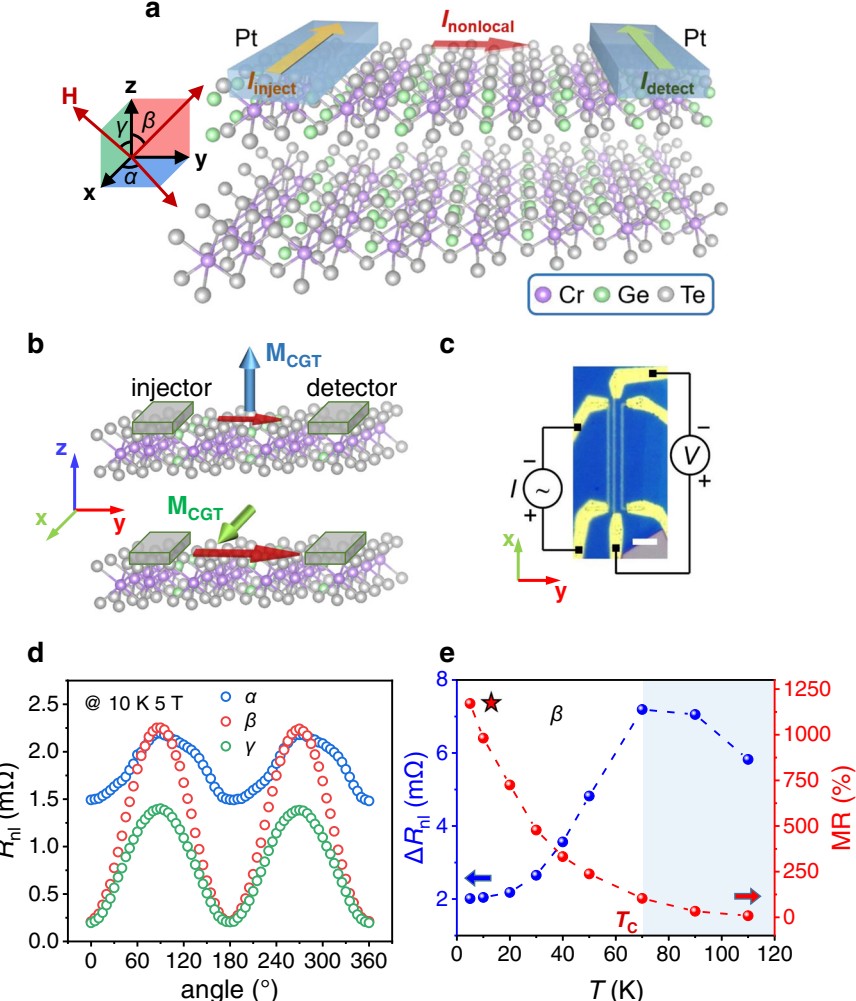

**Fig. 3 | Colossal nonlocal MR effect in CGT. a** Schematic of transport measurements in the lateral nonlocal device on vdW CGT. The inset shows the definition of azimuthal angles and directions of the measurement. **b** Schematic of the mechanism of the nonlocal MR effect. The red arrows denote the current flows between the injector and detector. **c** Optical microscope image of the device and schematic of the measurement setup. Scale bar: 5 μm. **d** Angle-dependent nonlocal resistance $R_{nl}$ as the magnetic field rotated in **xy** plane ($\alpha$, blue), **yz** plane ($\beta$, red) and **xz** plane ($\gamma$, green) at 10 K temperature with 5 T magnetic field. **e** Temperature-dependent nonlocal MR (red) and the range of nonlocal resistance ($\Delta R_{nl}$, blue) as the magnetic field rotated in **yz** plane ($\beta$).

decreased $\Delta E$ with the increase of layers (Supplementary Fig. 5). Additionally, the obvious anisotropy of the angular MR curves is observed with the value of MR changing more rapidly as the magnetization near out-of-plane direction ($\gamma = 0°$) than in-plane direction ($\gamma = 90°$). The measurements of temperature-dependent electrical resistivity can also further support the MB effect in CGT thin flakes (Supplementary Fig. 6).

### Nonlocal magnetoresistance effect

Based on the vdW MR in CGT, we present a lateral device of nonlocal configuration with the channel of CGT acting as a magneto switch to further promote the performance. The nonlocal devices have been utilized in semiconductors for the transport measurements of magnon[33–35] or spin[36–38], which is based on the mechanism of diffusive transport. The geometry of the non-local resistance measurement is shown schematically in Fig. 3a, in which two separated platinum strips function as the injector and detector on top of the CGT thin flake. As the longitudinal charge current is applied in the injector along **x** direction, a transverse nonlocal current will be induced in the CGT channel, diffusing towards the detector. The non-local current carries the longitudinal electric potential difference from the injector to the detector which generates the current in the detector, leading to a longitudinal nonlocal voltage or resistance (nonlocal voltage divided

by the applied current) along $x$ direction at the detector. By manipulating the band structure of CGT channel with the magnetization, the on-off control of the nonlocal current is achieved reflected by the change of nonlocal resistance ($R_{nl}$). As schematically displayed in Fig. 3b, with out-of-plane magnetization, the bands splitting causes the larger resistance of the CGT channel between two electrodes, which shuts the diffusion of nonlocal current leading to small $R_{nl}$. The situation of in-plane magnetization is just the opposite, which opens the channel with large nonlocal current and $R_{nl}$. Considering the diffusive transport mechanism at the semiconductor channel and its interface with the metallic electrodes, vdW lattice-induced MR effect has the opportunity to be further enhanced in the nonlocal configuration.

The experimental setup and the fabricated device are exhibited in Fig. 3c. A low-frequency AC current ($I_{inject}$) was applied to the injector strip of Pt with the thickness of 8 nm and width of 600 nm, while the nonlocal resistance ($R_{nl}$) was detected along the same direction as the injector. Figure 3d illustrates the typical angle-dependent $R_{nl}$ as the magnetic field rotated in **xy** plane ($\alpha$), **yz** plane ($\beta$) and **xz** plane ($\gamma$) at 10 K and 5 T magnetic field, which is large enough to fully align the magnetization (Supplementary Note 2). The results show stark difference with nonlocal magnon transport and directly suggest a distinct dominant mechanism from spin or magnon transport[33] in the

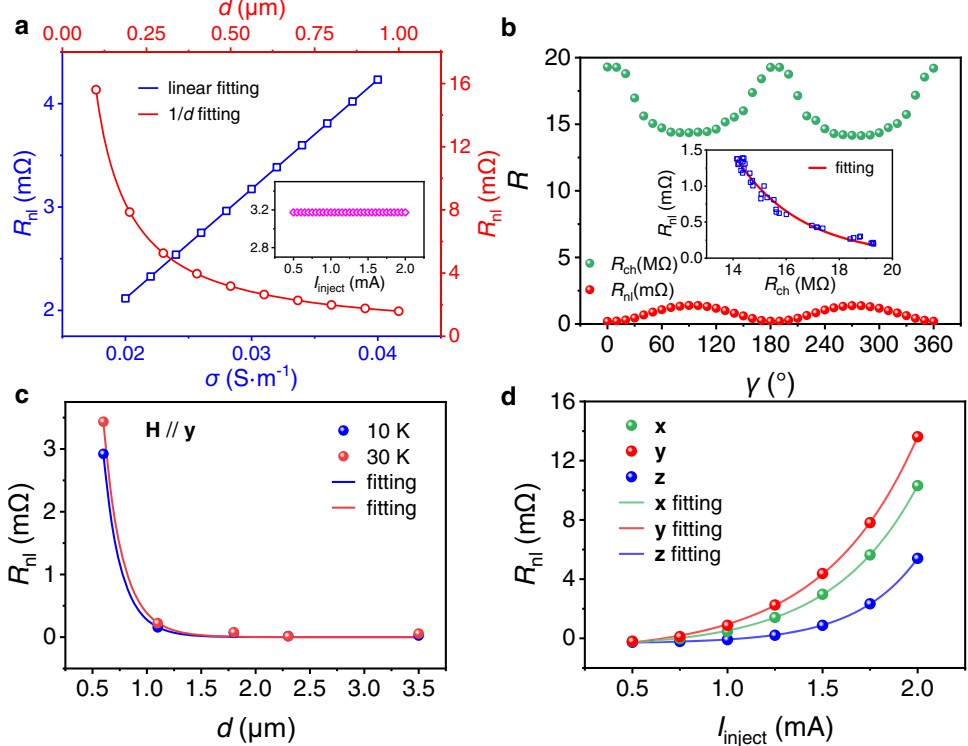

**Fig. 4 | Diffusive transport mechanism in nonlocal measurements. a** Simulated dependence of nonlocal resistance ($R_{nl}$) on the conductivity of the channel $\sigma$ (blue, bottom and left axes), the distance between two electrons $d$ (red, top and right axes) and $I_{inject}$ in the inset using the finite element analysis. The geometry of the device is the same as the experiments and the material of the channel is set to metallic material with similar conductivity as CGT. **b** Angle-dependent channel resistance ($R_{ch}$) and $R_{nl}$ as the magnetic field rotated in **xz** plane ($\gamma$). The inset shows the relevance between $R_{ch}$ and $R_{nl}$ extracted from the angle-dependent measurement. **c,d** Amplitude of $R_{nl}$ as a function of the channel width between the injector and detector (**c**) and the applied current in the injector (**d**).

device. Firstly, the observed change of $R_{nl}$ during the $\gamma$ scan should be undetectable in nonlocal magnon transport. In $\gamma$ scan, the magnetization (**M** in $xz$ plane) keeps perpendicular to the polarization of spin current injected by platinum ($\sigma$ // **y**), leading to the absence of nonlocal magnon transport or resistance change. Secondly, in $\alpha$ and $\beta$ scan, the polarity of angle-dependent $R_{nl}$ is opposite to the nonlocal magnon transport. Based on our measurement setup, due to the inverse Spin Hall effect, the low $R_{nl}$ should be obtained in nonlocal magnon transport when the magnetic moment is parallel to the spin polarization (**M** // $\sigma$ // **y**)[33]. In contrast, a high $R_{nl}$ is achieved in both $\alpha$ and $\beta$ scan. The negligible effect of nonlocal spin or magnon transport is further supported by the control experiment in which an electrode of tungsten with the spin Hall angle opposite to platinum[39,40] was utilized as the detector (Supplementary Note 3). Therefore, the dependence of $R_{nl}$ on the direction of magnetization is dominated by the MR effect and named as nonlocal MR. Note that the polarity of the angle-dependent $R_{nl}$ is exactly opposite to the local one in Fig. 2, which is consistent with the model proposed in Fig. 3b. Similarly, the $\alpha$ scan of $R_{nl}$ results from the in-plane AMR in which the channel resistance is maximum perpendicular to the nonlocal current and minimum parallel to that (Supplementary Fig. 7).

Most importantly, in line with our expectation, the nonlocal device shows good performance with the value of angular nonlocal MR defined as (max($R_{nl}$) − min($R_{nl}$)) / min($R_{nl}$) exceeds 1000 and 600% in $\beta$ and $\gamma$ scans, respectively. In the temperature-variation measurements, the value of nonlocal MR shows fast increase with the decrease of temperature as illustrated in Fig. 3e, which reaches 1200% in the $\beta$ scan at 5 K. This result reflects that when temperature decreases, the performance is promoted with the enhanced magnetism. In addition, the difference between $R_{nl}^y$ (maximum value) and $R_{nl}^z$ (minimum value), $\Delta R_{nl}$ first increases with the temperature up to 70 K and then decreases with the further increase of temperature. This shows the competition

between enhanced conductivity and diminished magnetism as the temperature increases. The critical temperature of 70 K is just around $T_C$ of CGT. The measured angular MR above $T_C$ is resulted from the paramagnetic state, which can also contribute to MR effect under large magnetic field.

The nonlinear dependence of $R_{nl}$ on the channel resistance ($R_{ch}$) and the excitation current ($I_{inject}$) can reflect the diffusive transport mechanism in the nonlocal device based on vdW semiconductor CGT. Firstly, for comparison, the situation of the nonlocal device based on the channel of a metallic material with similar conductivity as CGT was simulated through the finite element analysis. As illustrated in Fig. 4a, $R_{nl}$ is proportional to the conductivity ($\sigma$) of the channel and inversely proportional to the distance between two electrodes ($d$). According, $R_{nl}$ is inversely proportional to $R_{ch}$, which is $R_{nl}$ ~ 1/ $R_{ch}$. In addition, $R_{nl}$ shows no change with the $I_{inject}$ as illustrated in the inset. Therefore, based on the mechanism of simple electric conduction, although the resistance is still very large, the nonlocal device shows no effect on the enhancement of MR. In the situation of CGT, the tendency of variation of $R_{nl}$ is also opposite to $R_{ch}$ in both angle-dependent shown in Fig. 4b and temperature-dependent measurements (Supplementary Fig. 8), which is consistent with the mechanism of the nonlocal MR proposed in Fig. 3b. Differently, $R_{nl}$ shows fast decline as the increase of $R_{ch}$, $d$ and rapid promotion as the increase of $I_{inject}$, all with the exponential relation as shown in the inset of Fig. 4b, Figs. 4c and 4d receptively, which indicates a different mechanism from the simple electric conduction in a metallic channel. In addition, Fig. 4c illustrates the slow-down of the decline of $R_{nl}$ with $d$ as the temperature increases from 10 K to 30 K, resulting from the simultaneously decreased resistivity with enhanced electron mobility in the higher temperature. To sum up, according to exponential relationships in the nonlocal device based on the channel of semiconductor CGT, $R_{ch}$ acts as the barrier and $I_{inject}$ acts as the excitation in the diffusion of the nonlocal current in the

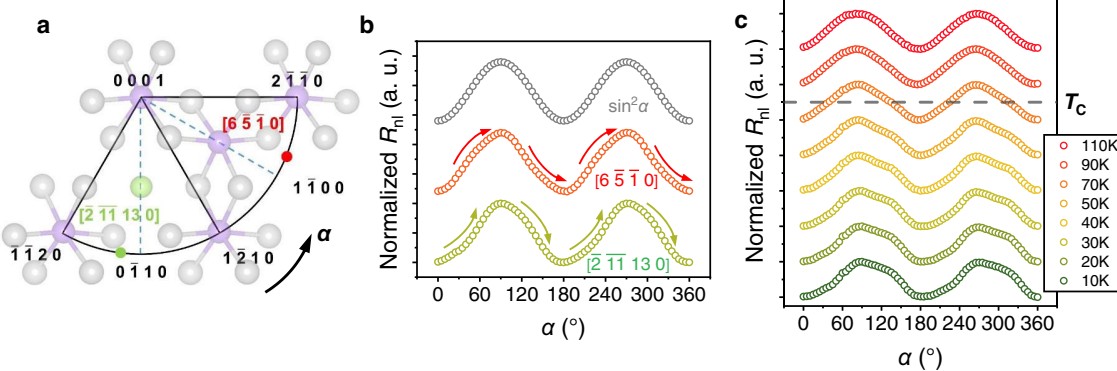

**Fig. 5 | Intralayer lattice anisotropy-induced modulation. a** Geometry of the intralayer anisotropy in CGT. **b** Normalized angle-dependent nonlocal resistance ($R_{nl}$) signal injector-detector strip along [65$\bar{1}$0] (red) and [$\bar{2}$ $\bar{1}$1 13 0] (green) crystal orientations. The standard signal of $\sin^2\alpha$ (grey) is also exhibited for comparison. **c**, Normalized angle-dependent $R_{nl}$ at selected temperature from 10 K to 110 K as the magnetic rotated in **xy** plane ($\alpha$).

channel. The nonlinearity in the diffusive transport process promotes the local MR effect and realizes the colossal nonlocal MR effect. Benefited from the enhanced MR effect, the nonlocal device also shows the applications in the detection of the magnitude and direction of the external magnetic field (Supplementary Fig. 9).

### Intralayer lattice-induced modulation

Besides the significant anisotropic lattice between interlayer and intralayer, the intralayer structure of honeycomb also exhibits anisotropy in CGT. As illustrated in Fig. 5a, there are two in-plane directions with high symmetry, which are <2$\bar{1}$$\bar{1}$0> and <1$\bar{1}$00 > . These two directions possess different atomic arrangements leading to the in-plane anisotropic exchange coupling with the strength along <2$\bar{1}$$\bar{1}$0> weaker than <1$\bar{1}$00>[41]. Consequently, the band structure will also be modulated with the magnetization along different in-plane orientations. According to the calculations, the top of the valance bands with **M** //<2$\bar{1}$$\bar{1}$0> is around 13 meV lower than **M** //<1$\bar{1}$00> away from the fermi level directly leading to enhanced resistance (Supplementary Fig. 10). This intralayer coupling between the spin and lattice can be reflected in the $\alpha$ scan and related to the crystalline directions.

Accordingly, two devices were fabricated with the electrode along different directions. As shown in Fig. 5b, the $\alpha$ scan shows opposite deviation in the two devices compared to the standard signal of $\sin^2\alpha$, which is consistent with the modulation of band structure. In the device along [$\bar{2}$ $\bar{1}$1 13 0], **M** crosses [1$\bar{2}$10] after rotating around 40 °, therefore, $R_{nl}$ shows a downward deviation around 45 ° due to the enhanced resistance of CGT. The situation in the device along [65$\bar{1}$0] is opposite with **M** across [10$\bar{1}$0] after rotating around 50 °. It is worth mentioning that the in-plane lattice anisotropy is much smaller and the resistance change only appears as the deviation on the basis of $\sin^2\alpha$ with the magnitude of ~10% of the total amplitude. Therefore, despite of the 60 ° period of the in-plane lattice, the deviation is only obvious at the extremums of the derivative of AMR where resistance changes fastest and leads to the significant deviation. In contrast, around the zeros of the derivative, the change of resistance is pretty slow with the negligible deviation. The similar principle is also verified by the devices on the same sample with different directions and the deviation is also reflected in the $\alpha$ scan of local resistance (Supplementary Note 4). Furthermore, the deviation gradually reduces with the increase of temperature and vanishes above $T_C$ of CGT as shown in Fig. 5c, which demonstrates its origination indeed related to the magnetic exchange coupling.

Based on the results above, we now summarize the principle of lattice-induced modulation in vdW CGT. The lattice of CGT induces the anisotropic exchange coupling along different crystal directions which is mainly bridged by the hybridization of Te-$p$ orbitals. The orbital momentum is consequently anisotropic which is larger in the direction with the weaker orbital hybridization. Meanwhile, the SOC in CGT is also mainly contributed by the Te atoms. Therefore, the anisotropic orbital momentum leads to the SOC field changing with the crystal directions. With the interaction between the SOC field and the magnetization, the magneto band structure is achieved. Specifically, the significant crystalline anisotropy between in-plane and out-of-plane directions results in the colossal MR effect and the in-plane anisotropy leads to the deviation.

In conclusion, we experimentally observe the vdW MR effect in thin flakes of vdW semiconductor CGT, which reveals the spin-modulated band structure originated from the anisotropic vdW lattice. Not only the significant anisotropy between interlayer and intralayer, but also the intralayer anisotropy between different orientations are reflected in MR measurements. The lateral nonlocal device with CGT as a magneto switch also shows promoted performance. Our work reveals the colossal MR effect induced by significant lattice anisotropy inherent to vdW semiconductor and shows the expectation of corresponding lateral nonlocal devices in the applications of spintronics.

## Methods

### Device fabrication and sample characterization

The CGT thin flakes were mechanically exfoliated from the crystal, transferred on Si/SiO₂ substrates using polydimethylsiloxane (PDMS) and directly capped with polymethylmethacrylate (PMMA) for the protection from oxidization. The Pt strips with Ti (10 nm)/Au (70 nm) were fabricated through electron-beam lithography and lift-off processes. The platinum was deposited in a magnetron sputtering system with the vacuum better than $1 \times 10^{-7}$ torr. The thickness of the flakes was measured by Atomic Force Microscope and the crystal orientations were measured by Electron Backscatter Diffraction.

### First-principle calculations

Our first principle calculations were performed using the Vienna ab initio simulation package (VASP)[42,43] with projector-augmented wave method[44,45]. Perdew-Burke-Ernzerhof (PBE) functional[46] was used to treat the exchange-correlation interaction and the plane-wave basis. The Gamma-centered k-point mesh of $9 \times 9 \times 1$ was used in all calculations with SOC. A vacuum layer larger than 15 Å was adopted in all calculations of thin films. DFT-D3[47] was used to properly treat the interlayer van der Waals interaction. The GGA + $U$ method was used to treat localized 3$d$ orbitals, the $U_{eff}$ is selected to be 2 eV for the 3$d$ orbitals of Cr according to the previous studies[48]. The model of CGT bilayer was utilized in the comparison of in-plane and out-of-plane magnetizations and the CGT monolayer in the exploration of in-plane anisotropy.

## Data availability

The data that support the plots within this Article and other findings of this study are available from the corresponding author upon reasonable request.

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

## Acknowledgements

This work was supported by the National Key R&D Program of China (Grant No. 2021YFB3601300), the National Natural Science Foundation of China (Grant Nos. 52225106 and 51871130), and the Natural Science Foundation of Beijing Municipality (Grant No. JQ20010). C.S. acknowledges the support of Beijing Innovation Center for Future Chip (ICFC), Tsinghua University.

## Author contributions

C.S. and F.P. led the project. W.Z. and C.S. proposed the study. W.Z. prepared the samples and carried out the measurements with the help from H.B. and T.G. W.Z. and C.S. wrote the manuscript. L.H. and H.B. reviewed and optimized the manuscript. All authors discussed the results and commented on the manuscript.

## Competing interests

The authors declare no competing interests.
