## [Peer Review File · Nature Communications]

Reviewers' Comments:

Reviewer #1:

Remarks to the Author:

External magnetic field could induce Zeeman spin splitting of electronic band structure. Yet, such a Zeeman spin splitting is tiny with only an order of magnitude of 0.1 meV under a field of 1 T. Besides the Zeeman effect, an applied magnetic field also enforces a reorientation of the spin direction, which leads to a significant change of electronic band structure of the order of 10 meV via spin-orbit coupling. This is called a magneto band-structure (MB) effect [Ref28]. The MB effect can make spin direction a very useful parameter to engineer the band structure and physical properties.

In their manuscript, Zhu et.al. demonstrated the MB effect by a combined study of magnetoresistance (MR) measurement and first principles density functional theory calculations of thin flakes of vdW semiconductor Cr₂Ge₂Te₆. They observed very impressive colossal local and non-local MR beyond 1000%. The experimental finding of the MB effect will pave a new avenue of spintronics. I therefore strongly recommend its publication on Nat. Comm. after the authors address the following points:

- i) As shown in the DFT calculated band structure of Fig1b, significant change occurs in the top of valence band, instead of the bottom of conduction band. It means that only hole-carrier could exhibit large MB effects. I would suggest the authors add Hall measurement to check the type of carriers.
- ii) The terminology of "spin-lattice coupling" might be not appropriate. The applied external magnetic field does not change the lattice properties such as crystal symmetry or phonon properties etc.
- iii) The authors claim that "...excludes the contribution of spin or magnon transport..." in Line 165. This is a very strong statement. I would suggest the authors provide a detailed analysis to support the argument, for example the comparison with [Ref 33].

Reviewer #2:

Remarks to the Author:

In this manuscript, the authors have observed a giant magnetoresistance in Cr₂Ge₂Te₆ thin flakes. They also show that this magnetoresistance exhibits highly anisotropic behavior and suggests this behavior is associated with a spin-lattice coupling induced magneto band structure effect. The experimental results are reasonable. However, I am unclear about the advantages of 2D vdW magneto compared to previous studies and bulk materials. It is necessary to clarify the superiority of vdW materials in order to be accepted by this journal.

Comments

1. Is the origin of the observed magnetoresistance essentially the same as the bulk case cited in Refs 30, 31, 32?
2. If the origin is the same, why is there a thickness dependence, as shown in Fig. 2?
3. If different, how are they different? For example, fig1 only shows the bilayer band calculation; can you compare it with the bulk band calculation?
4. In this paper, the origin of the resistance increase in in-plane M is attributed to the decrease in the spin majority band energy. However, the spin minority band energy is rising, and since this band is also below the Fermi level, this band should not be ignored. In this case, the effects of the majority and minority bands would be averaged out, and this type of magnetoresistance would not occur.
5. Can you demonstrate the effect of increasing the gap in other experiments? For example, the temperature dependence of electrical resistivity, carrier concentration, etc.

Minor Comment

The results of the band calculation in fig1 b should have a magnified view to show the change in the band at the Γ point.

Response Letter of NCOMMS-22-27238-T

We appreciate the positive evaluation of our manuscript by Reviewer 1# **“They observed very impressive colossal local and non-local MR beyond 1000%. The experimental finding of the MB effect will pave a new avenue of spintronics. I therefore strongly recommend its publication on Nat. Comm. after the authors address the following points.”** and Reviewer 2# **“The experimental results are reasonable.”** Meanwhile, they raised several comments and questions about the work. Their comments are helpful for our further improvements. We address the issues raised by them point by point below. Amendments of our revised manuscript are summarized below. The revised parts in our manuscript are highlighted.

The main modifications include:

1. We have measured the carrier type of CGT as recommended by Reviewer 1 and show the results in Supplementary Note 1.
2. Based on the comment of Reviewer 1, we revised our title and change the terminology of “spin-lattice coupling” in the manuscript.
3. More detailed analysis is added to distinguish the nonlocal resistance from magnon transport.
4. According to Reviewer 2, the calculations of thickness-dependent band structure and detailed discussions are provided to illustrate the difference with bulk materials and the superiority of 2D vdW magnets. The results are added in Supplementary Fig. 5.
5. Temperature-dependent resistivity is measured to support the change of band gap in Supplementary Fig. 6.

Response to Reviewer 1

External magnetic field could induce Zeeman spin splitting of electronic band structure. Yet, such a Zeeman spin splitting is tiny with only an order of magnitude of 0.1meV under a field of 1 T. Besides the Zeeman effect, an applied magnetic field also enforces a reorientation of the spin direction, which leads to a significant change of electronic band structure of the order of 10 meV via spin-orbit coupling. This is called a magneto band-structure (MB) effect [Ref28]. The MB effect can make spin direction a very useful parameter to engineer the band structure and physical properties.

In their manuscript, Zhu et.al. demonstrated the MB effect by a combined study of magnetoresistance (MR) measurement and first principles density functional

theory calculations of thin flakes of vdW semiconductor Cr₂Ge₂Te₆. They observed very impressive colossal local and non-local MR beyond 1000%. The experimental finding of the MB effect will pave a new avenue of spintronics. I therefore strongly recommend its publication on Nat. Comm. after the authors address the following points:

Response: We appreciate the positive evaluation from the Reviewer and revise our manuscript according to following comments.

Q1: As shown in the DFT calculated band structure of Fig1b, significant change occurs in the top of valence band, instead of the bottom of conduction band. It means that only hole-carrier could exhibit large MB effects. I would suggest the authors add Hall measurement to check the type of carriers.

Response: We really appreciate the comment of the Reviewer and investigate the carrier type of Cr₂Ge₂Te₆ (CGT) through the measurements of field effect curve and the Hall effect which both demonstrate the hole-type carrier in CGT. The results and measurements setup are added in the Supplementary Information. Supplementary Fig. 1a schematically shows the fabricated CGT field effect transistor (FET). The measured field effect curve in Supplementary Fig. 1b exhibits the feature of p-type FET, indicating the hole-carrier in CGT. The contact between the electrodes and CGT sample may lead to the small on-off ratio. The tiny leakage current and cyclicity (20 cycles) are also exhibited to reflect the validity of the result. The Hall effect of CGT which can also detect the carrier type of semiconductors was furtherly measured. Based on the measurement setup in Supplementary Fig. 1c and the result in Supplementary Fig. 1d, we can also judge that the carrier is hole-type in CGT. The above results can support the observed large MB effect through the measurement of magnetoresistance. We also supplement the discussion in Page 5 Line 1 “Consequently, with the hole-carrier in CGT (Supplementary Note 1), the modulation at the top of the valance bands will largely cause the change of resistance, leading to MR effect.”.

Supplementary Note 1. Carrier type of CGT

The carrier type of CGT is measured through the field effect curve and Hall effect, which both illustrates the hole-type carrier of CGT. Supplementary Fig. 1a schematically shows the fabricated CGT field effect transistor (FET).

The measured field effect curves shown in Supplementary Fig. 1b exhibits the feature of p-type FET, reflecting the hole-carrier in CGT. Both cyclicality (20 cycles) and leakage current are also shown. This is furtherly supported by the measurement of Hall effect in Supplementary Fig. 1d, whose setup is shown in Supplementary Fig. 1c.

Supplementary Figure 1. **a**, Schematic of CGT field effect transistor. **b**, Field effect curves measured at 300 K. The cyclicality (20 cycles) of source-drain current (I_{ds}) and leakage current (I_{leak}) are exhibited in grey and black, respectively. One typical curve is highlighted in red. **c**, Measurement setup of Hall effect. **d**, Hall effect of CGT measured at 300 K.

Q2: The terminology of “spin-lattice coupling” might be not appropriate. The applied external magnetic field does not change the lattice properties such as crystal symmetry or phonon properties etc.

Response: We agree with the Reviewer and change the terminology of “spin-lattice coupling induced” into “vdW lattice induced”, which can directly illustrate the mechanism. We revised the terminology in the title and all over the manuscript. The title is changed into “**Van der Waals lattice-induced colossal magnetoresistance in Cr₂Ge₂Te₆ thin flakes**”.

Q3: The authors claim that “...excludes the contribution of spin or magnon transport...” in Line 165. This is a very strong statement. I would suggest the authors

provide a detailed analysis to support the argument, for example the comparison with [Ref 33].

Response: We weaken the statement of “exclude” and provide a detailed analysis for the comparison with Ref. 33 in Page 9 Line 8 “The results show stark difference with nonlocal magnon transport and directly suggest a distinct dominant mechanism from spin or magnon transport³³ in the device. Firstly, the observed change of R_{nl} during the γ scan should be undetectable in nonlocal magnon transport. In γ scan, the magnetization (\mathbf{M} in xz plane) keeps perpendicular to the polarization of spin current injected by platinum ($\boldsymbol{\sigma} // y$), leading to the absence of nonlocal magnon transport or resistance change. Secondly, in α and β scan, the polarity of angle-dependent R_{nl} is opposite to the nonlocal magnon transport. Based on our measurement setup, due to the inverse Spin Hall effect, the low R_{nl} should be obtained in nonlocal magnon transport when the magnetic moment is parallel to the spin polarization ($\mathbf{M} // \boldsymbol{\sigma} // y$)³³. In contrast, a high R_{nl} is achieved in both α and β scan. The negligible effect of nonlocal spin or magnon transport is further supported by the control experiment in which an electrode of tungsten with the spin Hall angle opposite to platinum^{39,40} was utilized as the detector (Supplementary Note 3). Therefore, the dependence of R_{nl} on the direction of magnetization is dominated by the MR effect and named as nonlocal MR.” and Supplementary Note 3.

Supplementary Note 3. Angular nonlocal MR with tungsten detector

The control experiment with the injector of platinum and detector of tungsten was performed. The schematic of the measurement is shown in Supplementary Fig. 3a. Due to the opposite spin Hall angle between Pt and W, in the spin-related transport, the polarity of the signal will be inverted with W detector (Pt inject, W detect) compared to Pt (Pt inject, Pt detect). In contrast, as illustrated in Supplementary Fig. 3b, the results with W detector is similar to that of Pt shown in Fig. 3c in the main text. The high R_{nl} is obtained with $\mathbf{M} // y$ and $\mathbf{M} // x$ in β and γ scan, respectively. With $\mathbf{M} // z$, it shows the low R_{nl} . Therefore, the spin-related transport is negligible in our nonlocal devices.

Supplementary Figure 3. **a**, Schematic of the measurement with the definition of azimuthal angles and directions. **b**, Angle-dependent ΔR_{nl} ($R_{nl} - R_{nl}^z$) of the β and γ scans with the 5 T magnetic field at 10 K.

Response to Reviewer 2

In this manuscript, the authors have observed a giant magnetoresistance in Cr₂Ge₂Te₆ thin flakes. They also show that this magnetoresistance exhibits highly anisotropic behavior and suggests this behavior is associated with a spin-lattice coupling induced magneto band structure effect. The experimental results are reasonable. However, I am unclear about the advantages of 2D vdW magneto compared to previous studies and bulk materials. It is necessary to clarify the superiority of vdW materials in order to be accepted by this journal.

Response: We appreciate the evaluation of the Reviewer and clarify the superiority of vdW materials in the following response.

Q1: Is the origin of the observed magnetoresistance essentially the same as the bulk case cited in Refs 30, 31, 32?

If the origin is the same, why is there a thickness dependence, as shown in Fig. 2?

If different, how are they different? For example, fig1 only shows the bilayer band calculation; can you compare it with the bulk band calculation?

Response: The observed magnetoresistance is originated from **the intrinsic vdW layered structure** of 2D vdW CGT, which is different from the bulk crystal

cases. In the vdW case, the distinct atomic alignment between interlayer and intralayer leads to the anisotropic orbital hybridization and spin-orbit field, which results in the modulation of band structure with different spin orientation. Due to the origin of vdW structure, **this modulation of band structure is expected to be widely realized in other vdW magnets.**²⁹ In contrast, the modulation of band structure in the bulk cases is originated from the space-time inversion symmetry broken³² or topological nodal line³⁰, **which exists in specific materials.**

Ref. 29 *npj Quantum Mater.* **5**, 30 (2020)

Ref. 30 *Nature* **599**, 576–581 (2021)

Ref. 32 *Phys. Rev. B* **104**, 214419 (2021)

We calculated the band structure of monolayer and bulk CGT and add the comparison in Supplementary Fig. 5. Besides the bilayer in Fig. 1, both monolayer and bulk show the band split with out-of-plane magnetization. Nevertheless, the splitting energy shows a significant decrease with the increase of thickness from monolayer to bulk as shown in Supplementary Fig. 5c. **The thickness-dependent result exhibits the identical feature of vdW materials and reflects the origin of vdW layered structure, which is distinct from crystal materials.** These simulations can support the measured thickness dependence. In monolayer, without interlayer orbital hybridization, the hybridization mainly occurs intralayer and the difference of spin-orbit field between in-plane and out-of-plane directions is the most significant, leading to the largest splitting energy. With the increase of thickness, the anisotropic spin-orbit field and the splitting energy gradually decrease. In bulk, due to the difference between interlayer and intralayer coupling, the band splitting still exists but is largely weakened.

Besides the distinct mechanism, the observed magnetoresistance in 2D vdW magnets **possess the superiority** compared to bulk crystal materials. VdW magnetic thin flakes can be easily exfoliated from crystals and transferred **with the preservation of fine crystal structure at the same time**, which enables the existence of band structure modulation and magnetoresistance in thin samples and heterostructures. **Heterogenous thin films with high crystal quality are demanded in highly integrated practical devices.** In contrast, it is **much more difficult** to obtain the thin films of those bulk crystals and keep the special band structure unchanged at the

same time, which is caused by **the commonly reduced crystal quality** during the film preparation process. It becomes even harder especially during the construction of heterostructure. Consequently, without high crystal quality, the magnetoresistance induced by the modulation of band structure will be hard to obtain. This could also be one of the reasons why most works related to the precise investigation of band structure chose the bulk crystal materials rather than thin films. From this perspective, the 2D vdW magnets are more advantageous.

We also add discussions above in Page 3 Line 5 “Recently, the MB effect was observed in bulk crystal materials due to topological nodal-line bands^{30,31} and symmetry-broken correlated band reconstruction³² which results in colossal MR effect with different spin orientations. Compared to those bulk crystal materials, 2D vdW magnets possess the advantage of being easily exfoliated from crystals transferred with the preservation of fine crystal quality. This enables the efficient modulation of band structures by the magnetization even in thin flakes and heterostructures, which is demanded in highly integrated electronic devices. Furthermore, the origin of anisotropic atomic alignment inherent to vdW layered structure makes the MB effect promising to be widely observed in 2D vdW magnets.” and Page 5 Line 24 “Compared to the sample of 35 nm, the further enhancement in the thinner sample shows the characteristic of 2D materials and reflects its origin of the vdW layered structure. This is supported by the calculations of thickness-dependent band structure which shows the decreased ΔE with the increase of layers (Supplementary Fig. 5).”.

Supplementary Figure 5. a,b Calculated band structure of monolayer (a) and bulk (b) CGT with the out-of-plane magnetization. c, Comparison of splitting energy among monolayer, bilayer and bulk CGT.

Q2: In this paper, the origin of the resistance increase in in-plane M is attributed to the decrease in the spin majority band energy. However, the spin minority band energy is rising, and since this band is also below the Fermi level, this band should not be ignored. In this case, the effects of the majority and minority bands would be averaged out, and this type of magnetoresistance would not occur.

Response: In ferromagnets, the carrier is highly spin-polarized, especially under large magnetic field above saturation. In this case, the carrier with spin minority will be rapidly scattered and show little contribution to the electric conduction. In contrast, the carrier with spin majority mainly contributes to the resistance. Therefore, compared to the spin minority bands, the change of spin majority bands will dominate the magnetoresistance in ferromagnetic CGT. Meanwhile, the resistance increases in out-of-plane M with the decreased spin majority band energy. We have provided more detailed discussion in Page 5 Line 9 “In ferromagnetic CGT, the carrier is highly spin-polarized. The spin minority will be scattered rapidly and contribute little to the electric conduction. Consequently, compared to the in-plane M , the conductivity of semiconductor CGT, which is dominated by the spin majority, is reduced with larger resistance.”.

Q3: Can you demonstrate the effect of increasing the gap in other experiments? For example, the temperature dependence of electrical resistivity, carrier concentration, etc.

Response: We appreciate the advice of Reviewer which will support the MR effect. We obtained the temperature dependence of electrical resistivity with magnetization along out-of-plane direction (z) and in-plane direction (y) in Supplementary Fig. 6. The temperature dependence of resistivity shows the feature of semiconductor in both situations and the resistivity shows significant difference between them below T_C (around 70 K). With increasing temperature, the difference gradually decreases and disappears. In addition, due to the high resistance of CGT flakes, it is pretty hard to obtain the carrier concentration through Hall effect, especially at low temperature below T_C . This has also been reported in previous work (*Nat. Nanotechnol.* **13**, 554–559 (2018)) “**due to the**

highly resistive nature of the samples, we failed to extract the exact value of carrier density via Hall measurements”. We measured the Hall effect of CGT flakes at 300 K in Supplementary Fig. 1d, which has already reached large Hall resistance. We provide a description about the result in Page 6 Line 5 “The measurements of temperature-dependent electrical resistivity can also further support the MB effect in CGT thin flakes (Supplementary Fig. 6).”.

Supplementary Figure 6. Temperature-dependent channel resistivity with magnetization along in-plane (y) and out-of-plane (z) directions under 5 T applied magnetic field.

Supplementary Figure 1. **a**, Schematic of CGT field effect transistor. **b**, Field effect curve measured at 300 K. The cyclicity (20 cycles) of source-drain current (I_{ds}) and leakage current (I_{leak}) are exhibited in grey and black,

respectively. **c**, Measurement setup of Hall effect. **d**, Hall effect of CGT measured at 300 K.

Q4: The results of the band calculation in fig1 b should have a magnified view to show the change in the band at the Γ point.

Response: We add the magnified bands below fermi level in the inset of Fig. 1b.

Fig. 1 | Van der Waals lattice-induced magneto band-structure effect in CGT. **a**, Schematic of the band splitting induced by the interaction between the anisotropic crystalline field and the magnetization. The bands occupied by spin majority and minority are colored in blue and grey, respectively. **b**, Calculated band structures of CGT bilayer with the in-plane and out-of-plane magnetizations. The fermi level is set to zero. The bands of spin majority and minority with out-of-plane magnetization at the top of the valence bands are colored in black and grey, respectively. The magnified bands below fermi level are shown in the inset.

Reviewers' Comments:

Reviewer #1:

Remarks to the Author:

The reply is satisfactory, and I therefore recommend its publication on Nat. Comm.

Reviewer #2:

None